# Insights into the Composition and Antibacterial Activity of *Amomum tsao-ko* Essential Oils from Different Regions Based on GC-MS and GC-IMS

**DOI:** 10.3390/foods11101402

**Published:** 2022-05-12

**Authors:** Weidan Li, Junjie Li, Zhen Qin, Yang Wang, Pengyu Zhao, Haiyan Gao

**Affiliations:** 1School of Life Sciences, Shanghai University, Shanghai 200444, China; wendylee@shu.edu.cn (W.L.); ljjly@shu.edu.cn (J.L.); qin_zhen@shu.edu.cn (Z.Q.); 18723021@shu.edu.cn (P.Z.); 2School of Environmental and Chemical Engineering, Shanghai University, Shanghai 200444, China; carollwangy@shu.edu.cn

**Keywords:** *Amomun tsao-ko*, essential oil, composition, GC-MS, GC-IMS, antibacterial activity

## Abstract

Chemical components are one of the most significant traits and attributes of plant tissues, and lead to their different functions. In this study, the composition of *Amomun tsao-ko* essential oils (AEOs) from different regions was first determined by a combination of gas chromatography–mass spectrometry (GC-MS) and gas chromatography–ion mobility spectrometry (GC-IMS). In total, 141 compounds were identified, of which terpenes and aldehydes were the main groups. Orthogonal partial least square discriminant analysis (OPLS-DA) distinguished the samples from different regions clearly, and the main differences were terpenes, aldehydes, and esters. Meanwhile, AEOs showed strong antibacterial activity against *Staphylococcus aureus* (*S. aureus*), and the minimum inhibitory concentration (MIC) and minimum bactericidal concentration (MBC) reached 0.20 mg/mL and 0.39–0.78 mg/mL, respectively. From correlation analysis, 1,8-cineole, (E)-dec-2-enal, citral, α-pinene, and α-terpineol were determined to be the potential antibacterial compounds. This study provides the basis for the variety optimization of *A. tsao-ko* and its application as a natural food preservative.

## 1. Introduction

*Amomun tsao-ko* Crevost et Lemaire (*A. tsao-ko*) is a perennial herb belonging to the Zingiberaceae family and is a fragrant plant. Its dried fruit is a commercially crucial spice in Southeast Asia and is used in traditional Chinese medicine [1] as well. It is a kind of economic crop that grows in humid forests in tropical and subtropical areas [2]. *A. tsao-ko* is mainly distributed in China, Vietnam, Laos, and other Southeast Asian regions. In China, it is mostly located in Yunnan Province, with a planting area and yield of 90% [3]. Because of its distinctive spicy smell, *A. tsao-ko* is usually used as a spice to develop food flavor [4]. *A. tsao-ko* is also a common Chinese medicine used to treat malaria, abdominal pain, phlegm, vomiting, dyspepsia, and other diseases [5].

Essential oils (EOs), extracted from the fruits, roots, stems, leaves, and other parts of plants, are natural, volatile, and aromatic liquids [6]. EOs and their phyto-constituents have been shown to display a full range of biological activities, including antimicrobial, insecticidal, and antiviral activities [7]. At present, EOs are mainly used in the preservation of various foods (meat products, dairy products, fruits and vegetables) and for food flavoring in the food industry. In terms of preservation, oregano EO [8] and *Satureja horvatii* EO [9] could significantly inhibit the growth of Listeria monocytogenes in meat; treating peaches with microencapsulated clove EO not only increased their antioxidant properties, but also extended their shelf life by 5–6 days [10]; chitosan enriched with cinnamon EO could reduce the rot rate of sweet pepper, and increase the activity of antioxidant enzymes in the sweet pepper [11]; lemongrass EO can inhibit bacteria such as Escherichia coli and Salmonella in lemon, pear, and apple juice [12]; and in milk, lemongrass EO and its components can effectively inhibit gram-positive bacteria and improve its physical and chemical quality. Meanwhile, many EOs and their components classified as "Generally Recognized As Safe" (GRAS) by the United States Food and Drug Administration (FDA) are widely used in food additives [6]. As natural substances, there is a growing trend of using plant essential oils as safe preservatives in food [13]. *A. tsao-ko* essential oil (AEO) has been proven to have strong anti-bacterial and anti-fungal effects [14,15,16]. The edible and antimicrobial properties of *A. tsao-ko* make it a potential natural and environmentally friendly food preservative.

The composition of an EO determines its function, and it is very important to understand the components to predict the functional effects [17]. Many factors determine the chemical composition of EOs, including plant species, geographical origin, seasonal variation, and plant maturity. *A. tsao-ko* is still in the state of “semi-wild”, with mixed germplasm types and a lack of varieties [3], and the altitude suitable for *A. tsao-ko* growth is widely distributed (700–2000 m) [2]. Because current studies on the composition of AEO focused on a single area, it is of great significance to study the composition of *A. tsao-ko* essential oils (AEOs) from different regions. At present, the studies about the components of AEOs mainly focused on the effects of different pretreatment methods [18,19] of AEOs and the different physical properties of *A. tsao-ko* [20]; furthermore, only gas chromatography–mass spectrometry (GC-MS) has been used to study its composition [20,21,22]. Although GC-MS can analyze the specific compounds qualitatively and quantitatively using the excellent separation of gas chromatography (GC) and the high selectivity of mass spectrometry (MS) identification, some trace compounds cannot be detected because of their limited separation ability [23].

Gas chromatography–ion mobility spectrometry (GC-IMS) has emerged as an analytical method in the past few years. It is better suited for the identification of trace volatile organic compounds (VOCs) due to the differing mobility of gas-phase ions in a constant electric field. [23]. It also has been widely utilized in food taste analysis, quality inspection, and other fields because of its high sensitivity, fast analysis speed, and simple operation [23,24,25]. Compared with GC-MS, GC-IMS has more advantages in the detection of trace compounds, small molecular compounds, and efficient separation of isomers in the detection due to its powerful instrumentation capabilities. For the detection of essential oil samples, GC-MS and GC-IMS can complement each other [26].

Hence, our objective was to qualitatively and quantitatively analyze the composition of AEOs from three typical *A.tsao-ko*-producing areas in the Yunnan province of China using GC-MS and GC-IMS technology. The characteristics of the AEOs from each producing area were determined. Moreover, the antibacterial activity of AEOs against *Staphylococcus aureus* (*S. aureus*) was determined to obtain the potential antimicrobial substance by correlation analysis. This study will reveal a more comprehensive composition of AEO for further applications. It also offers a theoretical basis for the identification of *A. tsao-ko* from different producing areas, as well as the selection and breeding of excellent varieties of *A. tsao-ko*. It also plays a vital role in the development of food preservation using AEO.

## 2. Materials and Methods

### 2.1. Plant Materials and Chemicals

Yunnan province is split into five floristic zones based on distinct vegetation (Figure 1) [27], and *A. tsao-ko* is mainly distributed in three of them: the south and southwest (zone I), the southeast (zone II), and the west and northwest (zone IV) [21]. In this study, Wenshan (A) was in zone II, Nujiang (B) was in zone I, and Tengchong (C) was in zone IV. In these three regions, ripe *A. tsao-ko* fruits with approximate maturity were collected from November to December 2020. Three batches for each region were collected, and more detailed information is provided in Appendix A. All fruits were obtained by drying naturally to a constant weight.

N-alkanes ($C_{7}$–$C_{40}$) (Anpu Experimental Technology Co., Ltd, Shanghai, China) and n-ketones ($C_{4}$–$C_{9}$) (Sinopharm, Beijing, China) of chromatographic pure grade were applied as the external standard of GC-MS and GC-IMS, respectively. Methanol was of chromatographic pure grade from Sinopharm (Beijing, China), and other chemicals were analytical reagents from Sinopharm (Beijing, China).

### 2.2. Extraction of AEOs

With slight modifications, AEOs were extracted from the whole fruit of *A. tsao-ko* using the methodology of Sun et al. [15]. The whole *A. tsao-ko* fruit was immediately pulverized into a fine powder of approximately 40 mesh, and then soaked in water with a liquid–solid ratio of 8:1 for 0.5 h. The mixture was subjected to hydro-distillation in a clevenger-type facility for 4 h until the volume of AEOs remained constant. The essential oils were filtered via organic filters with a 0.22 µm pore size, collected in sealed dark-brown glass vials, and refrigerated at 4 ${}^{\circ }$C for further examination.

### 2.3. GC-MS Analysis

Methanol was used as solvent to dilute the oils 100-fold for GC-MS analysis. GCMS-TQ8050 NX (Shimadzu, Japan) equipped with an HP-5 capillary column (30 m × 0.25 mm × 0.25 µm, Agilent, Palo Alto, CA, USA) was applied to detect the compound of AEOs. The carrier gas (helium) was at a flow rate of 1 mL/min. The injector and MS transfer line temperatures were set at 250 ${}^{\circ }$C and 230 ${}^{\circ }$C, respectively. The oven program was as follows: the initial column temperature was sustained at 50 ${}^{\circ }$C for 2 min, increased to 130 ${}^{\circ }$C at a rate of 5 ${}^{\circ }$C/min, then increased to 190 ${}^{\circ }$C at a rate of 4 ${}^{\circ }$C/min for 2 min, and finally increased to 220 ${}^{\circ }$C for 5 min. The transfer line temperature was 240 ${}^{\circ }$C. The effluent from the capillary column was split at a ratio of 20:1 (*v*/*v*). Working parameters of the MS were set as follows: ionization energy, 70 eV; scan range at 30–550 *m*/*z*.

### 2.4. GC-IMS Analysis

GC-IMS analysis was performed utilizing the GC coupled with an ion mobility spectrometry instrument (Flavourspec®-G.A.S. Dortmund Company, Dortmund, Germany). In summary, 2 µL of AEO was injected into the headspace vial (20 mL) and incubated at 40 ${}^{\circ }$C for 10 min. Following that, 100 µL of the AEO’s headspace was injected at rate of 2 mL/min, while the temperature injector was held at 85 ${}^{\circ }$C. The gas chromatography was carried out via a capillary column (FS-SE-54-CB-1, 15 m × 0.53 mm × 1 µm, CS-Chromatographie Service GmbH, Langerwehe, Germany) maintained at 60 ${}^{\circ }$C with nitrogen as the carrier gas. With the initial flow rate (2.0 mL/min) remaining for 2 min, the rate of internal linearity was increased as follows: 10 mL/min within 8 min, 100 mL/min within 10 min, 50 mL/min within 10 min, and finally, 150 mL/min, which was sustained for 5 min. The total chromatographic separation time was 35 min.

### 2.5. Agar Disc Diffusion Assay

The antibacterial ability of the AEOs was determined applying the agar disc diffusion method. Briefly, a suspension (0.1 mL of $10^{8}$ CFU/mL) of *S. aureus* ATCC 6538 was spread on Luria Berstani (LB) agar plates. Each disc was placed at the center of the plate with 5 µL of AEO. These plates were incubated at 37 ${}^{\circ }$C for 24 h. Ultrapure water was used to provide negative controls. The diameter of the zones of inhibition (ZOI) against *S. aureus* ATCC 6538 was used to assess antibacterial efficacy. Each assay was carried out three times.

### 2.6. Minimum Inhibitory Concentration (MIC) and Minimum Bactericidal Concentration (MBC) Assay

With slight modifications, MIC and MBC against the *S. aureus* were determined using the procedure published by Diao et al. [28]. In short, 50.00 mg/mL essential oil solution with a mixture of 10/1000 (*v*/*v*) DMSO and 1/1000 (*v*/*v*) Tween-80 solution as the solvent was arranged [15]. Two-fold serial dilutions of AEOs were used in sterile LB broth medium, extending from 0.10 mg/mL to 50.00 mg/mL. To each tube, 100 µL of the solution and 100 µL of bacterial cells ($10^{6}$ CFU/mL) were added. The 100 µL of solvent, instead of the solution, was utilized as a positive control. The 100 µL of 0.85% saline, instead of the bacterial suspension, was utilized as a negative control. The tubes were incubated at 37 ${}^{\circ }$C for 24 h before being inspected for growth with a microplate reader set to 600 nm (SpectraMax M2, Molecular Devices, Sunnyvale, CA, USA). The MIC was determined as the lowest concentration of AEO that exhibited no obvious growth (absorbance difference ≤0.005).

Following the MBC determination, 100-fold dilutions using drug-free LB broth from each tube that showed no turbidity were incubated at 37 ${}^{\circ }$C for 24 h. The MBC was determined as the lowest concentration of AEO that exhibited no obvious growth throughout cultivation (absorbance difference ≤0.005). All of the trials were repeated threes times.

### 2.7. Statistical Analysis

Using the NIST 17 database, compounds in the GC-MS data were identified by comparing the linear retention indices and mass spectra. The GC-IMS data were examined using the special software including LAV (from G.A.S., Dortmund, Germany version 2.0.0), Reporter, Gallery Plot, and GC × IMS Library Search. With NIST Library and IMS database retrieval software from G.A.S., the detected VOCs were determined by combining the retention index (RI) and drift time (Dt).

All measurements were conducted three times, and the results are shown as the mean values with standard errors (SE). Significant differences and Spearman’s correlations were investigated by SPSS version 23 (Chicago, IL, USA). Orthogonal partial least square discriminant analysis (OPLS-DA) and Principal component analysis (PCA) were carried out by SIMCA version 14.1 (Umetrics, Umea, Sweden). The variable importance of projection (VIP) included in the OPLS-DA model was to describe the total relevance of the variable in the model. A schematic diagram to summarize the experimental design is provided in Figure 2.

## 3. Results and Discussion

### 3.1. Composition of AEOs by GC-MS

Detailed information about the compounds identified by GC-MS is displayed in Appendix A. The AEOs selected in this study contained a total of 115 compounds, which can be divided into six types: terpenes (67), aldehydes (19), alcohols (14), esters (6), ketones (5), and others (4). Terpenes (60.8–66.4%) and aldehydes (22.6–27.3%) were the main categories in the AEOs. Specifically, 20 substances (>1%), accounting for approximately 80% of the AEOs, were the main components (Table 1), all of which were terpenes and aldehydes. 1,8-cineole (28.9–36.6%) was the most abundant component, and (E)-dec-2-enal (8.5–12.0%), α-citral (2.1–6.9 %), β-citral (3.1–6.5%), α-methyl-benzenepropanal (3.9–5.5%), geranial dimethyl acetal (3.8–4.8%), (E)-dodec-2-enal (1.7–4.5%), α-terpineol (2.1–3.2%), (E)-oct-2-enal (2.0–2.9%), D-limonene (1.6–2.5%), and o-cymene (1.1–1.9%) were the other main components in all AEOs. A total ion current chromatogram of AEOs from three regions by GC-MS Is presented in Figure 3. According to Figure 3, the main components in Table 1 have been shown.

Terpenes are made from combinations of several isoprene units ($C_{5}$) and their oxygen-containing derivatives. According to the number of $C_{5}$ units of molecules, terpenes can be divided into monoterpenes, sesquiterpenes, diterpenes, and others; it is the main group of plant EOs [2]. Terpenes have various functional groups, including hydroxyl groups (linalool, geraniol, carveol, citronellol, and terpineol), aldehyde groups (citral and citronellol), carbonyl groups (carvone and camphor), ether bonds (1,8-cineole), and cyclic structures (cymene, pinene, limonene, and phellandrene) [29]. Therefore, it is a major constituent responsible for the biological properties of EOs. In this study, all terpenes were monoterpenes ($C_{10}$) and sesquiterpenes ($C_{15}$), and acyclic structures of mono-, bi-, or tricyclic monoterpenes were the majority (79.7%).

Our results for the main components were generally consistent with those of previous reports, but there were some differences in the content of some substances [30]. The contents of some aldehydes, such as (E)-dec-2-enal (8.5–12.0%), (E)-dodec-2-enal(1.7–4.5%), and (E)-oct-2-enal (2.0–2.9%), were generally higher than 3.0–6.1% [31], 0.29% [14], and 0.9–1.0% [16] of previously reported analyses, respectively. In addition, geranial dimethyl acetal (3.8–4.8%) and o-cymene (1.1–1.9%) were first discovered in AEO.

Many main compounds have been confirmed for use as fragrances in foods and beverages [32]. For example, 1,8-cineole has a mint-like and sweet odor; citral (α-citral and β-citral) has a lemon-like aroma; D-limonene has a lemon- and orange-like aroma; α-terpineol has an oil-, anise-, and mint-like odor; and (E)-dec-2-enal has a tallow-like aroma and can enhance umami palatability in soup [33,34]. (E)-oct-2-enal, used in preparing chicken, has a cucumber-like flavor [21]. 1,8-Cineole has strong antibacterial activity, especially against food-borne bacteria such as *S. aureus* and *Escherichia coli*, and anti-inflammatory activity as well [35]. (E)-dec-2-enal also has antimicrobial activity, and it has a strong pest-killing ability during grain storage [36]. Citral has good antimicrobial activity, and microencapsulated citral has also been widely studied [37]. The antimicrobial activities and applications of these components provide a solid foundation for the application of AEOs in food preservation.

### 3.2. Composition of AEOs by GC-IMS

The top-view plot (Figure 4A) of all AEOs was obtained by normalizing the ion migration time and reactive ion peak (RIP) position, where Dt and retention time (Rt) were depicted by the X- and Y-axes, respectively. Most of the signals were shown in the Rt scope of 100–1200 s and Dt of 1.0–2.0 a.u. To compare the differences between AEOs, the topographic plot A1 in Figure 4A was selected as a reference (Figure 4B). By comparing topographic maps, the contents of substances from the different producing areas are obviously different. The identification of each VOC was performed based on RI and Dt. Because of the high proton affinity or high concentration of the compounds, one analyte might produce multiple signals (protonated monomers, proton-bound dimers, trimers, and polymers) [38]. In total, 144 compounds were detected in the samples, of which 74 were identified (Appendix A). Among them, 25 produced multiple signals, including 21 dimers (D) and 4 polymers (P). For the 40 single VOCs, there were 14 aldehydes, 10 terpenes, 6 ketones, 5 esters, 3 alcohols, and 2 others.

Terpenes and aldehydes were also the main classes in GC-IMS. Compared with the GC-MS results, GC-IMS identified 27 new compounds, which were mainly concentrated on the small molecules of $C_{2}$–$C_{10}$. Overall, except for the relatively high content of 1,8-cineole and (E)-oct-2-enal, α-pinene (P), β-pinene (P), and α-terpineol (M) were the only substances with multiple polymer forms, indicating their high concentrations. In addition, α-phellandrene (M), (E)-hex-2-enal (D), (E)-dec-2-enal (D), hexanal (D), heptanal (D), α-terpineol (P), and linalool (P) were also present at high concentrations. These substances have been approved as food additives for food flavoring [32]. α-Pinene has a range of biological activities, such as antibacterial, anti-leishmanial, anti-inflammatory, and antioxidant properties [39]. Both α-pinene and α-terpineol have good antibacterial activity against gram-positive and gram-negative bacteria [39]. (E)-oct-2-enal and (E)-hex-2-enal also have antibacterial and antifungal activities, which are attributed to the existence of the α and β bonds [36]. Lanciotti et al. [40] discovered hexanal, (E)-hex-2-enal, and hexyl acetate cannot only extend the shelf life of “minimally processed foods”, but can also improve their hygienic safety.

Notably, there have been no reports of the $C_{2}$–$C_{5}$ substance in AEOs. The content of small molecular aldehydes such as (E)-hex-2-enal, hexanal, and heptanal also significantly increased compared with the low concentrations (0.01–0.13%), or was not reported in previous studies [14,20,30,31]. This is because GC-IMS is more suitable for the detection of small-molecule ($C_{2}$–$C_{10}$) volatile substances than GC-MS. This result can be complementary to the results of GC-MS, which provided a theoretical basis for comprehensively revealing the components of AEOs.

### 3.3. Analysis of Compounds Detected in AEOs from Different Regions

According to the significance analysis results of GC-MS (Appendix A), it can be found that there are some regularities in different regions, mainly focusing on terpenes, aldehydes, and esters. The content of monoterpene hydrocarbons, especially sabinene, (−)-β-pinene, β-myrcene, α-phellandrene, and 3-carene in sample A (Wenshan) was lower than that in other regions. Monoterpene aldehydes such as isocitral, (Z)-isocitral, and 7-methyl-3-methylene-6-octenal were significantly higher in sample B (Nujiang). The ingredients of some aldehydes, such as (E)-hex-2-enal, octanal, cis-4-decenal, and decanal were also significantly higher in sample B (Nujiang). Almost all sesquiterpenes were highest in sample C (Tengchong); α-muurolene, γ-muurolene, and α-nerolidol were significantly different; and di-epi-1,10-cubenol, β-cineol, and γ-cineol were detected only in sample C. In addition, sample C was characterized by esters. Geranyl acetate is a unique high-content substance of sample C; other esters also show higher concentrations in sample C, including acetic acid, octyl ester, [(E)-dec-2-enyl] acetate, benzoic acid, 4-(1,1-dimethylethyl)-, ethenyl ester, trans-2-dodecen-1-ol, acetate, and bornyl acetate.

Unlike GC-MS, the fingerprint of GC-IMS is more suitable for comparing the differences between different samples. The fingerprints were obtained from the signal intensities of all compounds in the topographic plot (Figure 5). Each row depicts a sample, and each column depicts a signal peak. It is notable that the signal intensities can only be compared among different samples and not different substances, because the sensitivity of the instrument to different substances varies greatly. Overall, the contents of most identified aldehydes and terpenes were consistent among different producing areas, but there were some differences among samples from the same producing area. This may be related to the similar phenotypic traits of *A. tsao-ko* in various producing areas. The change in topography and altitude causes the interlacement of vegetation types in various producing areas [1]. In addition, the classification of *A. tsao-ko* has not been determined, which results in unstable seed genetic characteristics and great variability [3].

An enlarged fingerprint with compounds with obvious content differences is shown in Figure 5B. Among the identified substances, the contents of nonanal and heptanal were the highest in sample A; decanal, (E)-dec-2-enal, and d-camphor were the highest in sample B; and β-pinene and 2-ethyl furan were the highest in sample C; and all these substances were significantly different (*p* < 0.05). Some unidentified compounds (75, 78, 81–84, 86, 99, 129, etc.) also showed obvious differences among the three regions. Notably, all esters were mainly concentrated in the samples from sample C, which is consistent with the above results. It is concluded that the AEOs of sample C (Tengchong) are characterized by esters. It was discovered that as *A. tsao-ko* was distributed from a producing area at lower altitude and/or latitude to a location of higher altitude and/or latitude, the fruit production of *A. tsao-ko* increased [41]. In our research, the content of many compounds in sample B1 with the highest altitude (2408 m) was visibly higher than that in the other samples. A3, with the highest latitude in A, showed the highest content of more compounds. These conclusions are congruent with the results of Yang et al. [41].

### 3.4. Multivariate Analysis of Compounds of AEOs from Different Regions Based on PCA and OPLS-DA

Because of the difficulty and inaccuracy of identifying the AEOs by direct visual inspection, analyzing the compounds of AEOs in a more progressed way was attempted. Multivariate analysis (MA) was utilized to investigate the relative variability within diverse varieties in this research, and it contained PCA and OPLS.

PCA is often used to classify many samples and reveal the relationship between variables by reducing the dimension or transforming multiple indicators into a few comprehensive indicators [38]. The relative content of GC-MS and the signal intensity of GC-IMS were used to set up the PCA model of compounds in the samples, which are shown in Figure 6A,B. The cumulative variance contribution rate of PC1 (59.0%) and PC2 (17.2%) in GC-MS was 76.2%, and that of PC1 (42.6%) and PC2 (32.1%) in GC-IMS was 74.7%. The results of the PCA charts showed that the three groups overlapped and could not be completely differentiated.

Due to instrumental drift, artifacts, and other experimental varieties, the focus of a PCA model which is based on the unsupervised clustering method is diverted to the systematic variety, unrelated to the scientific issue of interest [42]. Thus, a supervised OPLS-DA model was built to further discover the differences between the different AEOs in this study. The score plots for the OPLS-DA model are shown in Figure 6C,D. The $R^{2}$ Y value (0.779) and $Q^{2}$ value (0.647) shown in Figure 6C, and the $R^{2}$ Y value (0.981) and $Q^{2}$ value (0.835) shown in Figure 6D, all indicated an acceptable reproducibility and prediction capability of the OPLS-DA model. It is apparent that OPLS-DA could differentiate these samples from the three regions better than PCA. This indicated that the supervised method of OPLS-DA was more suitable for the discrimination of AEOs from different regions. In addition, GC-IMS was more effective for identifying *A. tsao-ko* from different production areas according to its components.

In the OPLS-DA model (Figure 6C,D), the VIP values for crucial compounds to differentiate the three regions were studied. Moreover, variables of VIP > 1 were considered to have greater impacts on the discrimination method [25]. In the model, 52 and 38 variables were figured for VIPs larger than 1 in GC-MS and GC-IMS, respectively (Appendix A). More importantly, the identified compounds with VIPs larger than 1.5 were selected, including a total of five aldehydes, four terpenes, and two esters. Combined with the results shown in Appendix A, we can obtain the characteristic substances of each producing area. Sample A is rich in (E)-dodec-2-enal (1.76) and heptanal (D) (1.74), whereas sample B is rich in (E)-hex-2-enal (D) (3.17), (E)-dec-2-enal (2.71), and (E)-oct-2-enal (D) (2.47). Sample C is still characterized by its high content of esters (trans-2-dodecen-1-ol, acetate (2.52), (E)-2-decenyl acetate (2.28)), and also some monoterpenes, including α-phellandrene (3.25), linalool (M) (1.60), and α-pinene (P) (1.52). There were also some unidentified substances (86, 83, 81, 82) that were different from the AEOs from the three producing areas, which was consistent with Figure 5B.

### 3.5. Antibacterial Activity of the AEOs from Different Regions

#### 3.5.1. Antibacterial Activity of the AEOs

*S. aureus* can produce staphylococcal enterotoxins, and the ingestion of polluted food can lead to food poisoning [29]. Thus, it poses a significant risk to food safety. We tested the disc diameters of inhibition zones (DDs) qualitatively. We used the broth dilution method to measure the MICs and MBCs of AEOs against *S. aureus* for a further accurate and quantitative analysis of their antibacterial ability. The results of the disc diffusion test were 83.33 ± 0.94, 84.00 ± 0.00, and 84.00 ± 0.00 mm (plate inner diameter is 84 mm) for A, B, and C, respectively. The MICs were 0.39 mg/mL, 0.78 mg/mL, and 0.20 mg/mL for A, B, and C, respectively. The MBCs were 0.39–0.78 mg/mL, 1.56 mg/mL, and 0.39–0.78 mg/mL for A, B, and C, respectively. The varying concentration ranges represent the results of three batches from one sample.

The results showed that *S. aureus* was highly sensitive to AEOs from all the regions. The MIC of C (0.20 mg/mL) was the best, and those of A and C (0.39–0.78 mg/mL) were the same. This result is consistent with previous studies [14,16] and the antimicrobial effect is significantly better than that of other spices, such as the essential oil of *Cinnamomum cassia* (2.5 mg/mL) [43], and the essential oil of zanthoxylum (10 mg/mL) [28].

#### 3.5.2. Main Antibacterial Substances Analysis of AEOs

The antibacterial effect of essential oils from different origins varies with composition. To gain further insight into the relationship of all detected components with an antibacterial effect, the correlation analysis of the contents of the compounds and the MIC results was performed using Spearman’s correlation method. This value represents the correlation between the substance and antibacterial activity. Substances with significant negative correlations are shown in Table 2, and all substances had extremely significant correlations (*p* < 0.01) or significant correlations (*p* < 0.05).

Most of these compounds were terpenes, which proved to be more related to antibacterial activity. The substances with higher contents (>1%) were α-nerolidol, α-terpineol, and (E)-dec-2-enal, and their VIP values were more than 1. Previous studies showed that α-nerolidol and α-terpineol can inhibit *S. aureus* [44,45], and the content of (E)-dec-2-enal was positively correlated with the antibacterial activity [46].

In the main components of the AEOs, it has been reported that the MIC of 1,8-cineole, citral (with nanostructured lipid carriers), α-pinene, and (E)-dec-2-enal against *S. aureus* reached 6 µg/mL [35], 125 µg/mL [37], 210 µg/mL [39], and 250 µg/mL [36], respectively. The cause of inhibition may be the destruction of the lipid part of the bacterial membrane to penetrate the bacteria [36]. Thus, the main substances against *S. aureus* in AEO may be 1,8-cineole, (E)-dec-2-enal, citral, α-pinene, α-nerolidol, and α-terpineol.

## 4. Conclusions

In total, 141 compounds were identified by the GC-MS and GC-IMS analyses of AEOs. These were composed of six categories: terpenes, aldehydes, alcohols, esters, ketones, and others. The primary components of the essential oils were 1,8-cineole, (E)-dec-2-enal, (E)-oct-2-enal, α-citral, β-citral, α-terpineol, α-pinene, and β-pinene. Small-molecule compounds below $C_{6}$ were first detected in AEO using GC-IMS. OPLS-DA could be applied to the *A. tsao-ko* classification from different regions. The main differences in the compounds of the oils from different producing areas were terpenes, aldehydes, and esters. In particular, AEOs from Tengchong are characterized by a high ester content. *S. aureus* is highly sensitive to AEO. The best MIC and MBC could reach 0.20 mg/mL and 0.39–0.78 mg/mL, respectively. A correlation analysis indicated that 1,8-cineole, (E)-dec-2-enal, α-citral, β-citral, α-pinene, and α-terpineol are likely antibacterial compounds. This study revealed the composition of *A. tsao-ko* and established an effective method to distinguish *A. tsao-ko* from different regions. These results provide a reference for variety optimization and the further development of natural food preservatives.

## Figures and Tables

**Figure 1 foods-11-01402-f001:**
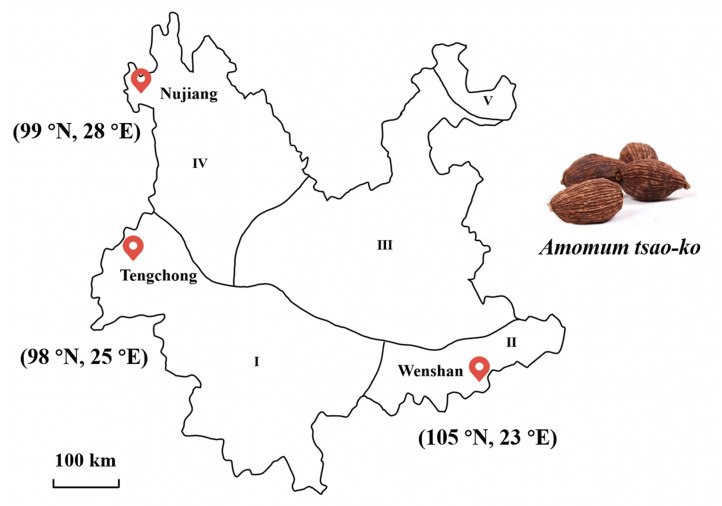
Floristic zoning map of Yunnan province and sampling sites of *Amomun tsao-ko* (*A. tsao-ko*). I (the south and southwest), II (the southeast), III (the middle plateau), IV (the west and northwest) and V (the northeast) represent different floristic zones of Yunnan province. The longitude and latitude of each region are labeled. A picture of *A. tsao-ko* (dried ripe fruit) is shown.

**Figure 2 foods-11-01402-f002:**
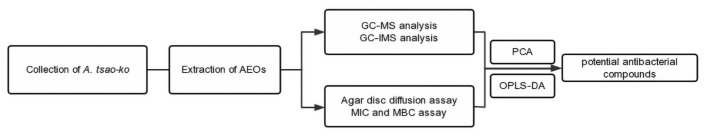
Schematic diagram of the experimental design.

**Figure 3 foods-11-01402-f003:**
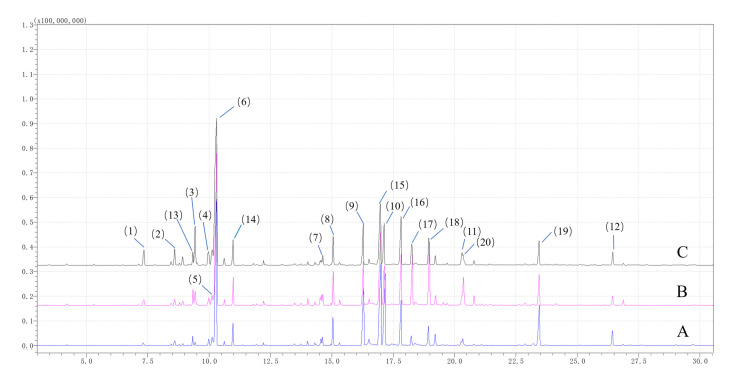
Total ion current chromatogram of *A. tsao-ko* essential oils (AEOs) from Wenshan (**A**), Nujiang (**B**), and Tengchong (**C**) by gas chromatography–mass spectrometry (GC-MS). Different colors represent different regions. Bule represents Wenshan (A), red represents Nujiang (B), and black represents Tengchong (C). The compounds with numbers correspond to the compounds listed in Table 1.

**Figure 4 foods-11-01402-f004:**
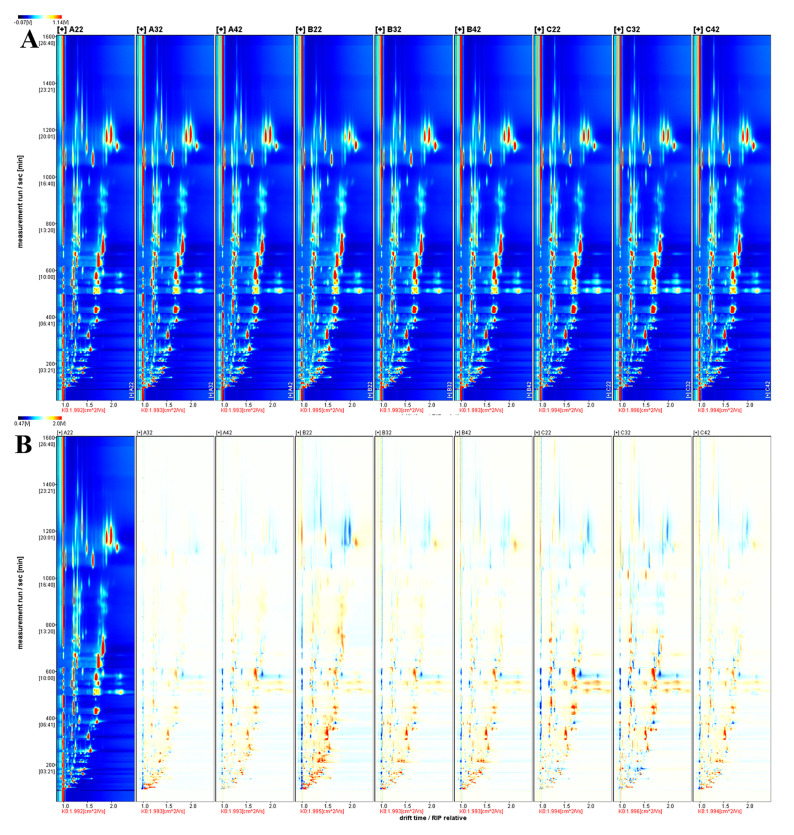
Topographic plots of the compounds isolated from AEOs from different regions using gas chromatography–ion mobility spectrometry (GC-IMS): A (Wenshan), B (Nujiang), C (Tengchong). (**A**) depicts the total volatile organic compounds (VOCs), and each colored point represents a compound. Red represents high concentration, and dark blue represents almost not present. Increasing color darkness from dark blue to red depicts increasing concentration. In (**B**), for VOCs of analytes with the same concentration compared to the reference (A1), the color would be white. Red and blue indicated that the concentration of VOC is more and less than the reference, respectively.

**Figure 5 foods-11-01402-f005:**
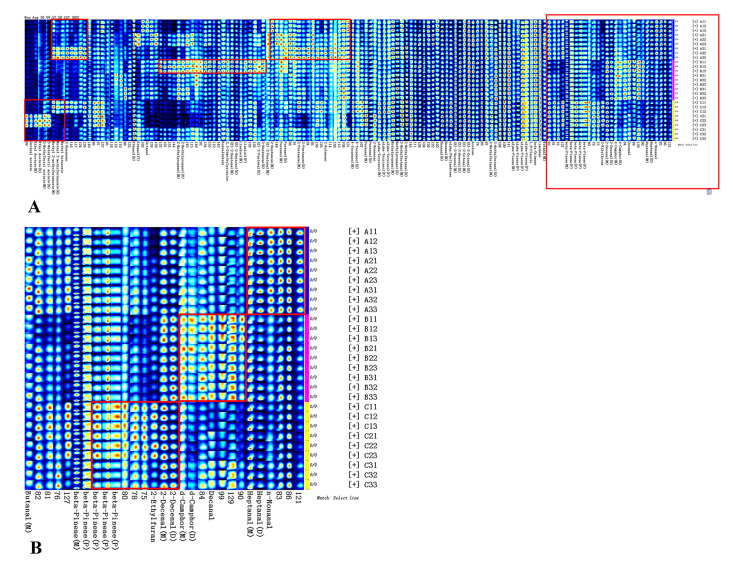
Fingerprints of the compounds isolated from AEOs from different regions using GC-IMS: A (Wenshan), B (Nujiang), C (Tengchong). (**A**) represents the fingerprint of all compounds detected by GC-IMS. (**B**) shows the enlarged fingerprint of compounds with obvious content differences among regions. Red represents high concentration, and dark blue represents almost not present. Increasing color darkness from dark blue to red depicts increasing concentration.

**Figure 6 foods-11-01402-f006:**
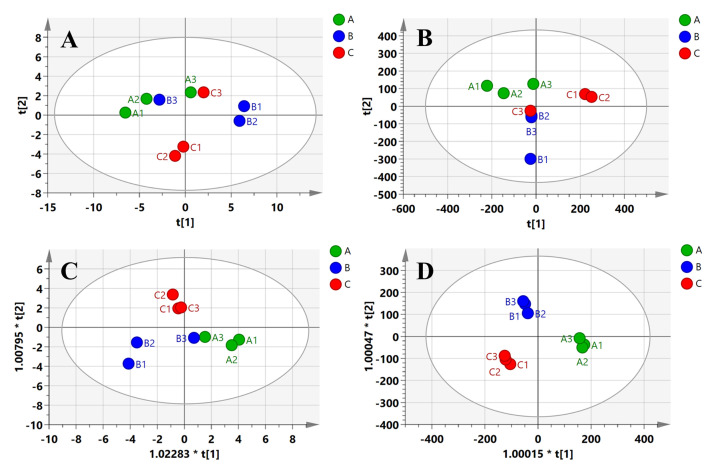
Scores plot for principal component analysis (PCA) and orthogonal partial least square discriminant analysis (OPLS–DA) models of compounds obtained from *A. tsao-ko* oils of different regions. (**A**,**B**) PCA models by GC–MS and GC–IMS; (**C**,**D**) OPLS–DA models by GC–MS and GC–IMS.

**Table 1 foods-11-01402-t001:** Main compounds in AEOs detected in gas chromatography–mass spectrometry (GC-MS).

No.	RI ^1^	Compound	MF	Concentration(%) *		
				A	B	C
		**Terpenes(12)**		**53.1**	**54.6**	**54.3**
1	932.5	α-pinene	$C_{10}H_{16}$	0.8 ± 0.2	1.0 ± 0.2	1.5 ± 0.2
2	976.6	(−)-β-pinene	$C_{10}H_{16}$	0.9 ± $0.1{\;}^{b}$	1.1 ± 0.3${\;}^{\mathrm{ab}}$	1.6 ± 0.3${\;}^{a}$
3	1005.9	α-phellandrene	$C_{10}H_{16}$	0.4 ± $0.3{\;}^{b}$	2.5 ± $0.6{\;}^{\mathrm{ab}}$	3.6 ± $0.6{\;}^{a}$
4	1024.0	o-cymene	$C_{10}H_{14}$	1.1 ± 0.1	1.9 ± 0.7	1.8 ± 0.7
5	1028.7	D-limonene	$C_{10}H_{16}$	1.6 ± 0.2	2.5 ± 0.8	2.1 ± 0.8
6	1035.4	1,8-cineole	$C_{10}H_{18}O$	28.9 ± 4.2	36.6 ± 6.5	29.1 ± 6.5
7	1179.6	(−)-4-terpineol	$C_{10}H_{18}O$	1.0 ± 0.2	0.7 ± 0.5	0.6 ± 0.5
8	1194.2	α-terpineol	$C_{10}H_{18}O$	3.0 ± 0.4	2.1 ± 1.5	3.2 ± 1.5
9	1237.1	β-citral	$C_{10}H_{16}O$	6.5 ± $1.5{\;}^{a}$	3.1 ± $1.3{\;}^{b}$	3.5 ± $1.3{\;}^{\mathrm{ab}}$
10	1267.1	α-citral	$C_{10}H_{16}O$	6.9 ± 2.8	2.1 ± 1.3	3.2 ± 1.3
11	1369.3	α-methyl-cinnamaldehyde	$C_{10}H_{10}O$	0.6 ± 0.6	0.5 ± 0.7	2.2 ± 0.7
12	1557.3	α-nerolidol	$C_{15}H_{26}O$	1.4 ± $0.2{\;}^{a}$	0.5 ± $0.4{\;}^{b}$	1.9 ± $0.0{\;}^{a}$
		**Aldehydes(7)**		**30.2**	**25.8**	**25.2**
13	1002.7	octanal	$C_{8}H_{16}O$	0.9 ± $0.1{\;}^{b}$	2.1 ± $0.7{\;}^{a}$	1.5 ± $0.7{\;}^{\mathrm{ab}}$
14	1057.2	(E)-oct-2-enal	$C_{8}H_{14}O$	2.0 ± 0.2	2.9 ± 0.8	2.1 ± 0.8
15	1262.2	(E)-dec-2-enal	$C_{10}H_{18}O$	12.0 ± 2.9	9.6 ± 3.1	8.5 ± 3.1
16	1291.6	α-methyl-benzenepropanal	$C_{10}H_{12}O$	5.5 ± 0.3	3.9 ± 1.9	5.3 ± 1.9
17	1328.0	geranial dimethyl acetal	$C_{12}H_{22}O_{2}$	3.9 ± 3.0	4.8 ± 2.0	3.8 ± 2.0
18	1336.1	indane-4-carboxaldehyde	$C_{10}H_{10}O$	1.4 ± $0.1{\;}^{a}$	0.8 ± $0.3{\;}^{b}$	1.2 ± $0.3{\;}^{\mathrm{ab}}$
19	1463.9	(E)-dodec-2-enal	$C_{12}H_{22}O$	4.5 ± 1.2	1.7 ± 1.5	2.8 ± 1.5
		**Ester(1)**		**-**	**-**	**2.1**
20	1373.4	geranyl acetate	$C_{12}H_{20}O_{2}$	-	-	2.1 ± 0.0

^1^ The value means the calculated RI. * Each value was indicated as mean ±standard deviation (SD) (*n* = 3) and
relatively quantitated by the peak area divided by the total peak area of each compound. “-” means not detected. ^a,b^ The different letters of the same compound show the significant difference among different regions (*p* < 0.05).

**Table 2 foods-11-01402-t002:** Spearman’s correlation between the compounds’ contents (Appendix A) from AEOs from different regions and the minimum inhibitory concentration (MIC) of *Staphylococcus aureus* (*S. aureus)*.

No.	Compound	Category	Correlation	*p*
1	elemol	Terpene	−0.953 **	0.000
2	α-muurolene	Terpene	−0.949 **	0.000
3	α-nerolidol	Terpene	−0.896 **	0.001
4	α-terpineol(M)	Terpene	−0.896 **	0.001
5	bornyl acetate	Ester	−0.845 **	0.004
6	2-methylbutanal(M)	Aldehyde	−0.843 **	0.004
7	[(E)-dec-2-enyl] acetate	Ester	−0.804 **	0.009
8	ethanol(D)	Alcohol	−0.791 *	0.011
9	γ-muurolene	Terpene	−0.791 *	0.011
10	4,10-epoxyamorphane	Terpene	−0.791 *	0.011
11	α-methyl-cinnamaldehyde	Terpene	−0.750 *	0.020
12	3-methylbutanal(M)	Aldehyde	−0.738 *	0.023
13	ethanol(T)	Alcohol	−0.738 *	0.023
14	ethyl acetate(M)	Ester	−0.685 *	0.042
15	(E)-dec-2-enal(M)	Aldehyde	−0.685 *	0.042
16	2-ethylfuran	Other	−0.685 *	0.042

“*” and “**” indicate significant difference at the 0.05 level and 0.01 level, respectively. “M”, “D”, and “T” mean protonated monomers, and proton-bound dimers and trimers detected by GC-IMS, respectively.

## Data Availability

Not applicable.

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
