# Peer review of "Insights into the Composition and Antibacterial Activity of Amomum tsao-ko Essential Oils from Different Regions Based on GC-MS and GC-IMS"

_foods, 2022, doi:10.3390/foods11101402_

Round 1

Reviewer 1 Report

The articles entitled „Insights into the composition and antibacterial activity of Amomum tsao-ko essential oils from different regions based on GC-MS and GC-IMS” is very complex and interesting, however is based on results obtained from common known and explore platn material.

Th authors should highlight more points regarding new elements in the study of a fairly well-known plant material.

Methodology, equipment are of very high quality, however there are a lot of mistakes in the text.

What is more in title there is “antimicrobial actvity”, but the authors determine this activity only on one organism.

Staphylolcoccus auresu is a latin name of microorganism and should be written in italic letters (line 8, line 107, 270 etc…)

Latin name Cinnamomum cassia – should written also in italic

There isn’t (in material and methods chapter) how the authors identified volatiles of interest.

What is more the accuracy of GC-method is based on the first decimal place. Please check the (%) in Table 1 and .

Line 136: the authors claim that they can divided volatiles into 6 groups: terpene, aldehyde etc.  Terpenes are group of compounds both hydrogenated terpenes and oxygenated derivatives. The aldehydes, acohols etc….. belongs to oxygenated, while terpenes should be named as hydrogenated terpenes.

Thera are also many errors of identified compounds names, especially in systematic one and some strangers like:

  • There are no specific name of the isomer like alpha, beta, gamma in many points. There are many names that start with a hyphen.

line 10 (-pinene, - terpinene. Specific ismers of these compounds)

line 179, 180, 181, 194,195, 196-201, 294, 297, most of compounds in Table 1 and 2, lina 304, 311

Thera are some mistakes during identification of volatiles. In table 2 muurolene 1 is identified two times – its imposible. The same remark for ethanol. What is ethanol D?

Bornyl acetate (Table 2) is ester not hydrogenated terpene.

What is “3Aldehydes (7)” in table 1?

Whats happed in 137 line and in line 242?

Line 184 : according to IUPAC the name of (E )-2-octenal and (E )-2-hexenal should  be (E )-oct-2-enal and (E )-2-hex-2-enal

These are only the most noticeable mistakes. Despite the high-quality research study, there are cardinal errors in the manuscript.

Reviewer 2 Report

The manuscript investigates the composition and antibacterial activity of Amomum tsao-ko fruit essential oil. The work is quite interesting and well-organized. I only have some observations that are listed in the following lines.

  1. There are several typos and grammatical errors. The work would benefit from close editing.
  2. The scientific names should be italicized or underlined throughout the manuscript.
  3. Add the source and purity of reagents and chemicals to the Materials section.
  4. Line 74: what was the water to plant material ratio? What was the rate of distillation? Did the oil volume remain constant after 4 hours of distillation?
  5. Which solvent was used to dilute the oils for GCMS analysis and at what ratio?
  6. Please double-check the units: Line 89: 2L or 2mL? Line 91: 100 L or 100 mL? Line 102: 5 L or 5 mL? Line 111: 100 L or 100 mL? Line 113: “The 100μL of 0.85The MBC was determined”?
  7. Add a schematic diagram to summarize the experimental design.
  8. Add a list of abbreviations.
  9. Figure 2: Provide a publishable version.

Reviewer 3 Report

The manuscript reports on the use of essential oil derived from a ginger species grown in China as a potential food preservative given the numerous antibacterial and antioxidant compounds that were identified using GC/MS and GC/IMS. At the first, the study is of interest for the Foods journal readers and can be characterized as a good contribution to the field.

However, there are numerous technical and grammar errors that must be revised thoroughly. For this purpose, I have indicated within the attached pdf the corrections that should be done including scientific additions. In addition, the authors must provide a typical gas chromatogram indicating the compounds of interest.

Furthermore, a brief dicussion about the advantages and limitations of GC/IMS analysis must be given in Introduction section compared to the conventionally used GC/MS analysis on the basis of the cost of consumables.

Based on these comments, I suggest a major revision.

Reviewer 4 Report

The article entitled " Insights into the composition and antibacterial activity of Amomum tsao-ko essential oils from different regions based on GC-MS and GC-IMS" presents an interesting characterization and use of the essential oil of Amomum tsao-ko Crevost et Lemaire; the work carried out can provide sustainable alternatives for the use of traditional fruits. However, it has many typographical errors, syntax errors, and disorder in presenting the information that does not facilitate the reading of the text.

In the following, I give a detailed revision of the manuscript.

  1. All scientific names must be in italics
  2. Did the authors use 2 liters of each sample for GC-IMS analysis? (Line 89)
  3. Please correct the number to CFU (Line 100)
  4. Did the authors put 5 L of AEO in the diffusion disc for the microbiological test?
  5. Did the authors use 100 L of bacterial cells for the MIC test?
  6. It was evaluated if the amounts of DMSO and Tween 80 did not present antimicrobial activity by themselves?
  7. What do the authors mean by the phrase "The 100 μL of 0.85The MBC was determined as follows:"?
  8. I consider it essential to locate the Figures close to where they are cited for the first time in the text.

It is complicated to understand the article results since the way they are presented is very disorganized; it is necessary to adjust all the details and inaccuracies that the manuscript presents so that it can be published

Round 2

Reviewer 1 Report

Dear Authors,

I accept the corrections made by the authors of the work, with the exception of one important point.

I completely disagree with the inclusion of systematic names of common known volatiles in the text of manuscript. If a compound has a common, well-known and recognizable name, it should be written in the manuscript instead of long, often incomprehensible systematic names.

for example lines: 10-12, table 1 and in many lines in text.

1,3,3-trimethyl-2-oxabicyclo[2.2.2]octane, (2E)-3,7-dimethylocta-2,6-dienal, (2Z)-3,7-dimethylocta-2,6-dienal, 2,6,6-trimethylbicyclo[3.1.1]hept-2-ene and 2-(4-methylcyclohex-3-en-1-yl)propan-2-ol

In the first review, I suggested to correct common names of compounds - not to change into the systematic names. This common names were very often without specifying isomerism.

Systematic name correction should be applied to compounds like octenal, decenal, dodecenal etc.

What is more: line 202 - the author have mention of S3 table, I can't see such a table in manuscript.

Author Response

Point 1: In the first review, I suggested to correct common names of compounds-not to change into the systematic names. This common names were very often without specifying isomerism. Systematic name correction should be applied to compounds like octenal, decenal, dodecenal etc.

Response 1: Thank you for this suggestion. We have changed systematic name of compounds to common names in the revised manuscript.

Point 2: What is more: line 202 - the author have mention of S3 table, I can't see such a table in manuscript.

Response 1: We appreciate the reviewer’s concerns. Due to article space limitations, we put the Table S3 in a file named “supplementary files”.

Reviewer 3 Report

The revised version has been totally improved. There is no need to give them with decimal points. Round them to the nearest value; i.e. 1356,7= 1357

Author Response

Point 1: The revised version has been totally improved. There is no need to give them with decimal points. Round them to the nearest value; i.e. 1356,7= 1357

Response 1: Thanks for the reviewer’s suggestion. We're sorry about that we haven't changed this issue. The reason is that the accuracy of GC-method is based on the first decimal place, so we didn’t round them to the nearest value.

Reviewer 4 Report

The article has improved considerably after the adjustments made by the authors; however, the authors continue without using the format of the journal, and the organization of the article is a bit deficient. Please, it is necessary to adjust the presentation of the work since, at this moment, this is harming the quality and content of the work carried out.

Author Response

Point 1: The article has improved considerably after the adjustments made by the authors; however, the authors continue without using the format of the journal, and the organization of the article is a bit deficient. Please, it is necessary to adjust the presentation of the work since, at this moment, this is harming the quality and content of the work carried out.

Response 1: We appreciate the reviewer’s concerns. We have further checked and revised the format of the manuscript, and we have added a graphical abstract to the submission system to make it easier to understand the organization of the article.
